# Microbial Relationship of Carious Deciduous Molars and Adjacent First Permanent Molars

**DOI:** 10.3390/microorganisms11102461

**Published:** 2023-09-30

**Authors:** Weihua Shi, Jing Tian, He Xu, Man Qin

**Affiliations:** 1Department of Pediatric Dentistry, Peking University School and Hospital of Stomatology & National Center of Stomatology & National Clinical Research Center for Oral Diseases & National Engineering Research Center of Oral Biomaterials and Digital Medical Devices & Beijing Key Laboratory of Digital Stomatology & Research Center of Engineering and Technology for Computerized Dentistry Ministry of Health & NMPA Key Laboratory for Dental Materials, Beijing 100081, China; shi_weihua25@163.com (W.S.); tianjing8905@163.com (J.T.); kqxuhe2004@126.com (H.X.); 2Department of Stomatology, Beijing Children’s Hospital, Capital Medical University, National Center for Children’s Health, Beijing 100045, China

**Keywords:** dental caries, microbiota, dentition, mixed, dental plaque, metagenome, sequence analysis

## Abstract

(1) Epidemiological studies have shown that deciduous molar caries are related to and more severe than permanent molar caries. This study aimed to investigate whether caries subtypes in deciduous molars were associated with caries in first permanent molars and to explore taxonomic and functional profiles of the microbiota involved in different subtypes. (2) 42 mixed-dentition children were recruited and were divided into DMC (carious deciduous molars but caries-free first permanent molars; *n* = 14), C (carious deciduous and first permanent molars; *n* = 13), and control (*n* = 15) groups. Metagenomic sequencing was performed for supragingival plaque samples obtained separately from deciduous and first permanent molars. (3) The microbiota of deciduous molars in the DMC and C groups differed not only in species-based beta diversity but also in compositional and functional profiles. In the C group-like subtype, 14 caries-related species and potential pathways were identified that could be responsible for the caries relationship between the deciduous and permanent molars. In the DMC group-like subtype, the overall functional structure, the levels of *Leptotrichia wadei*, *Streptococcus anginosus*, and *Stomatobaculum longum* and KOs in sugar transporters and fermentation, quorum sensing, and TCA cycle in their first permanent molars surprisingly resembled those of the C group rather than the control group. This suggested that these clinically sound first permanent molars were at a greater risk for caries. (4) Classification of deciduous molar caries according to the microbiota could serve as a caries risk predictor for adjacent first permanent molars.

## 1. Introduction

Cross-sectional and longitudinal epidemiological surveys have demonstrated that deciduous molar caries are a risk factor for caries formation in the first permanent molars [1,2,3,4]. However, the prevalence of caries in deciduous molars has been reported as 50%, much higher than that in the first permanent molars, which is only 6.3–12.3% by the age of 7–8 years [2]. Honkala et al. [3] also demonstrated that not all children with deciduous molar caries developed caries on their first permanent molars. The question then arises that among the children with caries in their deciduous molars, why do some develop first permanent molar caries soon after eruption, while others remain caries-free?

Clinical subtypes of caries have been investigated in the past few years [5,6]. Gormley et al. [5] classified early childhood caries (ECC) into five subtypes based on tooth surface-level caries and found that baseline ECC subtypes were associated with subsequent caries in both deciduous and permanent teeth. Accordingly, we wonder if there are subtypes of deciduous molar caries that may influence caries formation in adjacent first permanent molars. If so, the microbial etiology that causes caries in both kinds of molars, such as the caries-related microbiome (“Who are they?”) and its function (“What are they capable of doing?”) [7], requires investigation. We previously demonstrated that the microbial composition of plaque of newly erupted first permanent molars was distinct from that of deciduous molars in healthy mixed-dentition children [8]. Although several studies have identified caries-related taxa in the deciduous dentition [9,10,11,12], this does not explain the microbial etiology of the first permanent molar caries in mixed dentition. More work is needed to understand the microbial etiology of the association between the deciduous molar caries and the first permanent molar caries. On the other hand, we would also like to investigate whether the microbiota of sound first permanent molars adjacent to carious deciduous molars is also “healthy”.

In the present study, we used a metagenomic shotgun sequencing method to examine the supragingival plaque microbiota of deciduous molars and the first permanent molars in the mixed-dentition stage. The study aimed to investigate whether the proposed subtypes of carious deciduous molars were associated with different caries outcomes in adjacent first permanent molars and identify the taxonomic and functional profiles of the microbiota involved in different subtypes.

## 2. Materials and Methods

### 2.1. Ethics Statement

The study protocol was approved by the Institutional Review Board of Peking University Hospital of Stomatology (PKUSSIRB-201519003). Written informed consent was obtained from the parents or legal caregivers of all participants prior to enrollment.

### 2.2. Study Population and Sample Collection

Subjects were recruited from the Department of Pediatric Dentistry, Peking University School and Hospital of Stomatology and Wulutong Primary School in Beijing, China. Mixed-dentition-stage children who had four first permanent molars with fully erupted occlusal surfaces, at least four erupted permanent incisors, and retained deciduous molars were recruited. The exclusion criteria were: (i) systemic or infectious diseases, (ii) visually detectable enamel or dentin hypoplasia, (iii) dental restorations or pit and fissure sealants, (iv) oral appliances, (v) remaining tooth surfaces of the deciduous and first permanent molars that did not meet the sampling requirements, and (vi) antibiotic use or fluoride varnish application within the past 3 months.

Based on the caries status of deciduous and first permanent molars, the subjects were classified into three groups (Table 1): (i) the “healthy” (H) group, including children with caries-free dentition; (ii) the “deciduous molar caries” (DMC) group, including children with carious deciduous molars but caries-free adjacent first permanent molars; and (iii) the “caries” (C) group, with carious deciduous and adjacent first permanent molars. A total of 42 children aged 6.42–11.75 years were recruited, including 15, 14, and 13 children in the H, DMC, and C groups, respectively.

Plaque samples were collected from 10:00–11:30 a.m. All participants were instructed to avoid food and drink for 2 h and rinse their mouths before sample collection. Children were examined sitting with their backs towards the examiner and then lying down on the examiner’s legs with the examiner seated behind the child’s head. The dental examinations were performed using visual-tactile methods by the same pediatric dentist with the aid of headlamps and sterilized examination instruments. The kappa value for intra-examiner agreement in the diagnosis of caries was 0.834. A clerk assisted and recorded the basic information, dentition status, and dental caries. Caries were charted using the International Caries Detection and Assessment System (ICDAS) [13]. ICDAS codes 0–2 were defined as caries-free, and ICDAS codes 3–6 were defined as caries [14,15]. The caries status of the participants was measured using the decay, missing, and filling tooth surfaces indices in both deciduous teeth (dmfs index) and permanent teeth (DMFS index). Two samples of supragingival plaque were collected separately from the deciduous molars and first permanent molars for each participant. In the H group, all molars in the four quadrants were sampled. In the DMC group, the quadrants where two kinds of molars were both caries-free were excluded. For the C group, we only sampled the quadrants where deciduous and first molars both had caries. Sound deciduous molars in the sampling quadrants were also excluded. The sampling site of supragingival plaque was isolated from saliva using cotton rolls. Each sample was pooled from the sound smooth surfaces (ICDAS 1–2 surfaces were excluded) of corresponding teeth using dental excavators. Plaque samples were collected into 1.5 mL sterile centrifuge tubes (Axygen, CA, USA) containing 1 mL of TE buffer (10 mM Tris-HCl, 1 mM EDTA; pH 8) and placed on ice immediately after sampling. The dental examination and sampling procedures were generally completed within 15 min. Samples were then transported to the laboratory within 2 h and stored at −80 °C.

### 2.3. Shotgun Sequencing, Quality Control, Taxonomy, and Function Annotation

Supragingival plaque samples were sent on ice to Beijing QuantiHealth Technology Co., Ltd. (Beijing, China) for DNA extraction and metagenomic sequencing. Genomic DNA from samples was isolated and purified using an Oral Genomic DNA Extraction Kit (Takegene^®^, Ningbo, China) according to the manufacturer’s instructions. Extracted DNA quality was evaluated by electrophoresis in 1% agarose gels. DNA concentration was measured using a Qubit^®^ 3.0 fluorometer (Thermo Fisher Scientific, Waltham, MA, USA).

DNA libraries were prepared according to the vendor’s protocols and sequenced using the Illumina Novaseq 6000 or Hiseq X10 platform (Illumina, San Diego, CA, USA). To obtain high-quality clean reads, raw data were quality controlled as follows: (i) sequencing adapters were removed using the Cutadapt software (v1.14, parameter: -m 30) [16], (ii) reads with a quality score <20 or length <30 bp were removed using the SolexaQA package [17] of MOCAT2 software (v1.0), and (iii) contaminated host reads were removed using the SOAPaligner (v2.21, parameter: -M 4 -L 30 -V 10) [18]. The clean reads were then assembled using the SOAPdenovo software (v2.04, parameter: -d 1, -M 3, -L 500) [19], and the scaffolding contigs with lengths >500 bp were obtained. Based on these scaffolding contigs, the genes were predicted using MetaGeneMark [20]. A nonredundant gene catalogue was established using the CD-HIT (parameter: -c 0.95, -aS 0.9) [21]. The relative abundances of bacteria from different taxonomic levels were obtained by comparing clean reads with clade-specific markers using MetaPhlAn2 [22]. For functional analysis, the nonredundant gene catalogue was annotated using the eggNOG-mapper v2 [23]. Relative abundances of KEGG orthologs (KOs) were calculated based on gene abundances. The profiles of KEGG levels were generated based on KO abundances and the KEGG database.

### 2.4. Statistical Analysis

Statistical analyses were performed using the R software (version 4.1.2; the R foundation, Vienna, Austria). Beta-diversity was estimated using species/KO-based Bray-Curtis distance in the “vegan” package. Principal coordinate analysis was performed to visualize similarities among the microbial structures using the “ggplot2” and “ape” packages. The nonparametric method Adonis (*n* = 999) was used to compare the community structure among groups. The Kruskal-Wallis test was used to examine differences among three groups, and all *p*-values were adjusted for multiple testing (*q* value) using the Benjamini and Hochberg false discovery rate controlling procedure (“agricolae” package). *p* or *q* < 0.05 was considered statistically significant.

### 2.5. Data Availability

The datasets analyzed in this study can be found in the NCBI Sequence Read Archive (SRA) database under BioProject PRJNA913442.

## 3. Results

### 3.1. Study Population and Sequencing Data

The demographic and clinical characteristics of the participants are shown in Table 1 and Appendix A. There were no significant age or gender differences among the three groups. The C group had greater dmfs scores for deciduous molars and deciduous teeth compared to the DMC group.

Metagenomic sequencing of 84 samples generated 460.39 Gb (1.24–15.36 Gb) raw bases with an average of 5.48 ± 3.05 Gb per sample. After filtering 7.10 ± 6.79% sequences as the host gene sequences and quality control, high-quality clean reads for downstream analysis were obtained (Appendix A). Overall, 17 phyla, 29 classes, 46 orders, 90 families, 161 genera, 403 species, and 4818 KOs were detected.

### 3.2. DMC and C Group Deciduous Molars Harbored Related but Distinct Microbial Communities

To test the proposed caries subtypes in deciduous molars, we compared the plaque microbiota of deciduous molars between the DMC and C groups.

Variations were detected in species composition (taxonomic beta diversity), but not in functional beta diversity, between the deciduous molars of the DMC and C groups. Besides, taxonomic and functional beta diversity differences were all apparent when the DMC and C group deciduous molars were compared separately to the H group (Figure 1A,B).

The microbiota of the deciduous molars in DMC and C groups had similar compositions and functional changes compared to group H. For example, the relative abundances of *Streptococcus mutans* and *Leptotrichia wadei* were significantly higher in the DMC and C group deciduous molars, whereas, *Neisseria macacae* and *Capnocytophaga gingivalis* were enriched in the H group deciduous molars (Figure 1C). The functional analysis showed that phosphotransferase systems (PTS), which catalyze maltose/glucose (K02791), lactose (K02786), and fructose (K02769 and K02770) uptake, were more abundant in carious deciduous molars (DMC and C groups). Similar trends were also seen for pyruvate formate-lyase (PFL, K00656), a key enzyme in anaerobic sugar fermentation [24], and oligopeptide permease (Opp, K10823), which is a transporter for the quorum sensing (QS) system [25]. Enzymes in the tricarboxylic acid (TCA) cycle (K00024, K00382 and K00658), however, were enriched in the H group (Figure 1D).

Most notably, we found that the microbiota of deciduous molars in the C group differed from that in the DMC group not only in taxonomic composition but also in functional profiles (Figure 2). Compared to the DMC group, the relative abundance of *Streptococcus sanguinis* was significantly lower in the C group deciduous molars, while the relative abundances of *Prevotella nigrescens*, *Leuconostoc mesenteroides*, *Bifidobacterium longum*, *Bifidobacterium breve*, and *Bifidobacterium adolescentis* were markedly higher (Figure 2A, Appendix A). The KEGG mapper tool was used to reconstruct pathways from 186 differentially distributed KOs between deciduous molars from the DMC and C groups (Appendix A). KOs were finally annotated to 135 pathways; most pathways had KOs enriched in the DMC as well as in the C group (except for pathways in Human Diseases, Figure 2B). The most striking was map03010 ribosome, which had the most counts of KO (25 KOs), and all were less abundant in the C group deciduous molars (Figure 2B,C). It was also noteworthy that relatively large counts of KO (5 KOs, more importantly, were all enriched in the deciduous of C group) were linked to map00130 ubiquinone and terpenoid-quinone biosynthesis, particularly to menaquinone biosynthesis (Figure 2B,D).

### 3.3. Microbial Profiles of Children with Carious Deciduous and First Permanent Molars

In the C group, first permanent molar caries were influenced by the caries of adjacent deciduous molars. We compared the microbiota of deciduous and first permanent molars in this group to those in the H group and investigated the overlaps of differentially abundant taxonomic and functional profiles between the two tooth locations.

In the C and H groups, the vast majority of the sequences (>99%) belonged to one of seven phyla (Figure 3A): Actinobacteria (the average relative abundance: 24.93%), Bacteroidetes (20.41%), Proteobacteria (17.72%), Fusobacteria (14.17%), Firmicutes (13.55%), Candidatus Saccharibacteria (7.79%), and Spirochaetes (1.27%). At the species level, C and H groups had 29 and 21 differentially abundant species in the deciduous and permanent molars, respectively (Appendix A). The overlap between the two tooth locations included 14 species, which were all enriched in the C group (Figure 3B). In addition to the traditional cariogenic species, *S. mutans*, the identified species also included *L. wadei*, *Prevotella oulorum*, *Dialister invisus*, *Streptococcus anginosus*, *B. longum*, *Bacteroides fragilis*, *L. mesenteroides*, and *B. breve*.

Figure 3C shows the overall functional profiles in the C and H groups. About 50% of microbial function belonged to the KEGG level 1 pathway, metabolism. Carbohydrate metabolism (average relative abundance: 14.02%), amino acid metabolism (9.24%), signal transduction (5.8%), and membrane transport (5.21%) were the most abundant level 2 pathways. Deciduous molars had more differentially abundant KOs between the C and H groups (313 KOs) than the first permanent molars (243 KOs). The overlap of the two sites (51 KOs) were further investigated (Appendix A). In contrast to the corresponding tooth locations in the H group, PFL and Opp (K10823) were enriched in the deciduous and first permanent molars in the C group. Malate dehydrogenase (K00024) and dihydrolipoamide succinyltransferase (K00658) in the TCA cycle, however, showed enrichment in healthy molars, as did ATP-binding cassette transporters of arginine/ornithine (K10025, Figure 3D). Based on these findings, we proposed potential pathways for oral microbiota that may be responsible for caries in the C group (Figure 3E): (i) greater acid production from anaerobic fermentation, (ii) active communication via the QS system that could affect the acidogenicity and aciduricity of the oral pathogens, (iii) generation of fewer alkalis due to reduced arginine intake, and (iv) less active aerobic metabolism of sugar in caries process.

### 3.4. Microbial Composition and Functional Profiles of First Permanent Molars in the DMC Group

We investigated the microbiota of first permanent molars in the DMC group and found no variations in the microbial composition compared to the H and C groups (Figure 4A). At the KO level, surprisingly, the microbiota of the DMC group first permanent molars was functionally similar to the C group (*p* = 0.548) and not to the H group (*p* = 0.044, Figure 4B).

The first permanent molars in the DMC group had 25 (6.2%) differentially abundant species compared to the H group (Appendix A), including *L. wadei*, *Stomatobaculum longum*, *S. anginosus*, *Streptococcus parasanguinis*, and *Bacteroides dorei*, which were significantly enriched in the DMC group (Figure 4C).

Functionally, there were 424 differently distributed KOs between first permanent molars of the DMC and H groups (Appendix A), which could be mapped to 173 KEGG pathways. Compared to healthy controls, the transporters of various sugars, including maltose/glucose (K02791), lactose (K02786 and K02788), mannose (K02793, K02794, K02795 and K02796), fructose (K11196 and K02771), galactose (K20112), and cellobiose/diacetylchitobiose (K02759, K02760 and K02761), were more abundant in the DMC group in addition to PFL (K00656) and Opp (K10823, K15582 and K15583). Meanwhile, K00024, K00245, and K00658 in the TCA cycle were enriched in the first permanent molars of the H group (Figure 4D).

## 4. Discussion

This study tested the hypothesis that caries of deciduous molars could be assigned to subtypes based on their microbiota, which would be associated with differences in caries experience in their newly erupted first permanent molars. In our results, it was not surprising that the plaque microbiota of the deciduous molars of the DMC and C groups had similar caries-related changes compared to the normal microbiota (Figure 1C,D). More significantly, differences between the two groups were detected on both taxonomic and functional analyses. Compared to the DMC group, deciduous molars of the C group lacked the presence of *S. sanguinis*, which is considered to have an inverse relationship with dental caries [26]. Meanwhile, the deciduous molars of the C group had more abundant *P. nigrescens*, which was related to caries [27,28] and *L. mesenteroides*, which have similar acidogenic and aciduric characters like *S. mutans* [29]. Several *Bifidobacterium* species, known to have acidogenic and aciduric properties [30] and to be associated with caries in children [31,32], were only detected in the deciduous molars of the C group. We speculate that the overabundance of cariogenic species and the lack of health-related species made the microbiota in the deciduous molars of C group more cariogenic compared to the DMC group; this may increase the risk of caries for the adjacent first permanent molars.

Differences in abundances of certain metabolic pathways, including the ribosome pathway and menaquinone biosynthesis, were observed between deciduous molars of the two groups. Ribosomes are involved in protein synthesis and protect mRNA during bacterial stress responses [33]. Menaquinone, a component of the electron transport chain in Gram-positive aerobes and most anaerobes, plays a vital role in cellular respiration and oxidative phosphorylation [34]. Microorganisms in the DMC group deciduous molars had more abundant ribosomal proteins but less enzymes required for menaquinone synthesis compared to the C group. We assumed that the microbes in the DMC group may have had a greater ability to maintain mRNA stability and grow slower in response to environmental changes. The deciduous molars of the C group, however, lacked this stress response, and only the acidogenic and aciduric species could survive. To sum up, our results showed that the subtypes of carious deciduous molars in children based on the microbiota would be helpful in understanding the microbial etiology of caries and precision risk assessment for the first permanent molars.

In the C group-like subtype, we identified 14 species that may explain the relationship between deciduous and first permanent molar caries. The detection of *S. mutans* was not surprising. However, in agreement with previous studies [11,35], *S. mutans* accounted for <1% of the microbiota even in the C group. This indicates that other species may be responsible for caries development in this relationship. *L. wadei*, which is highly saccharolytic [36] and has been linked to caries in children [37,38], was also detected. Zhang et al. [11] suggested that the *Leptotrichia* species may be used for caries risk prediction in children with low levels of *S. mutans*. Several studies have reported that the difficult-to-culture *D. invisus* was more abundant in children with caries [9,12,39], but its metabolic properties remain unknown. Another candidate species in our study was *S. anginosus*, with acidogenic and aciduric properties similar to *S. mutans* [29,40]. Tanner et al. [39] reported that *S. anginosus* was associated with severe ECC at the community level.

Our results also enabled us to propose a model of function for caries development in this relationship (Figure 3E). PFL is a vital enzyme in anaerobic sugar fermentation [24], especially when sugar supply is limited or pH falls below neutral [41]. Abundant PFL in the C group molars indicated that plaque microbes in carious molars ferment sugars more actively than normal. In agreement with previous studies [12], KOs related to the TCA cycle were enriched in the H group, indicating that the aerobic metabolism of sugars was more active in healthy plaque. OPPs are necessary transporters in the ComRS QS pathway in *S. mutans* [25]. Microbiota of C group molars had more abundant OPPs, suggesting that the QS system functioned optimally in carious molars that might affect its biofilm formation, acidogenicity, and aciduricity [42]. Recent studies have also emphasized the beneficial effects of arginine in dental caries [43]. Reduced arginine intake in the C group may have led to the lack of protective effects. All these factors combined to shift the remineralization/demineralization balance towards the net loss of minerals and resultant caries.

In the other DMC group-like subtype, their first permanent molars were clinically caries-free, but we found that their microbial composition had changed. The levels of some caries-related microbes, including *L. wadei*, *S. anginosus*, and *S. longum*, were comparable to those in the C group. Anaerobic *S. longum* ferments sugars and produces acids. The growth remains below pH 5.5 [44]. Existing literature confirmed the association between *S. longum* and caries in children 3–8 years old [45] and adolescents [46]. In functional analysis, the overall functional structure of the first permanent molars in the DMC group was unexpectedly more similar to the C group rather than the H group. The DMC group’s first permanent molars also had enhanced potential for transportation and fermentation of sugars and QS but less active aerobic metabolism of sugars, indicating that the microbiota had switched towards a carious state functionally, even though they were still clinically normal.

There were several limitations of the study, for example, the small sample size; a lack of separating the teeth with a matrix band when sampling; the fact that the cross-sectional design of the study does not allow for conclusions such as that the higher abundance of cariogenic species in the deciduous molars do not necessarily mean the child will develop caries in the permanent molar; the relatively large age span; the lack of information about known risk factors for caries that could have affected the microbial composition of the dental plaque such as diet, exposure to fluoride, and socioeconomic status. Further work, such as a longitudinal study with a much larger sample size, is needed to reveal the pathogenic mechanism and signal transduction pathways associated with the relationship between carious deciduous molars and adjacent first permanent molars.

## 5. Conclusions

Our study tested the hypothesis that caries of deciduous molars could be assigned to subtypes based on their microbiota, which would be associated with differences in caries experience in their newly erupted first permanent molars. We also identified the species and pathways that may be responsible for this relationship. Changes in microbial composition and function were also noted in sound first permanent molars adjacent to carious deciduous molars, which place these clinically sound first permanent molars at an increased risk for caries.

## Figures and Tables

**Figure 1 microorganisms-11-02461-f001:**
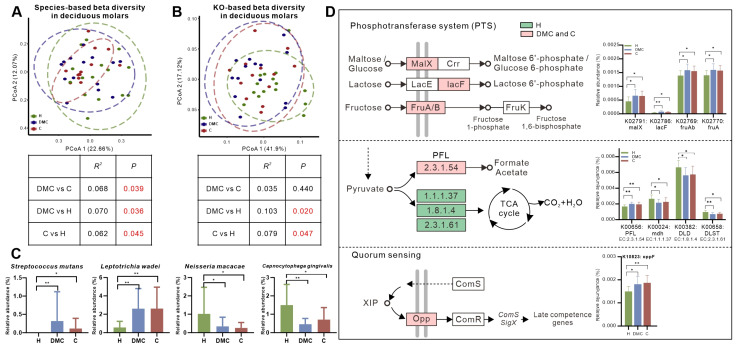
Similar microbial changes in the deciduous molars of the DMC and C groups. (**A**,**B**). Overall microbial taxonomic and functional beta diversity of the deciduous molars in three groups. Principal coordinate analysis was conducted based on species- (**A**) and KO-level (**B**) Bray-Curtis distance; the nonparametric method Adonis (*n* = 999) was used to examine community differences among groups. Red text indicates *p* < 0.05. (**C**,**D**). Similar microbial compositional and functional changes of deciduous molars in the DMC and C groups compared to controls. * *q* < 0.05, ** *q* < 0.01. Pathways in D were reconstructed using the KEGG mapper tool.

**Figure 2 microorganisms-11-02461-f002:**
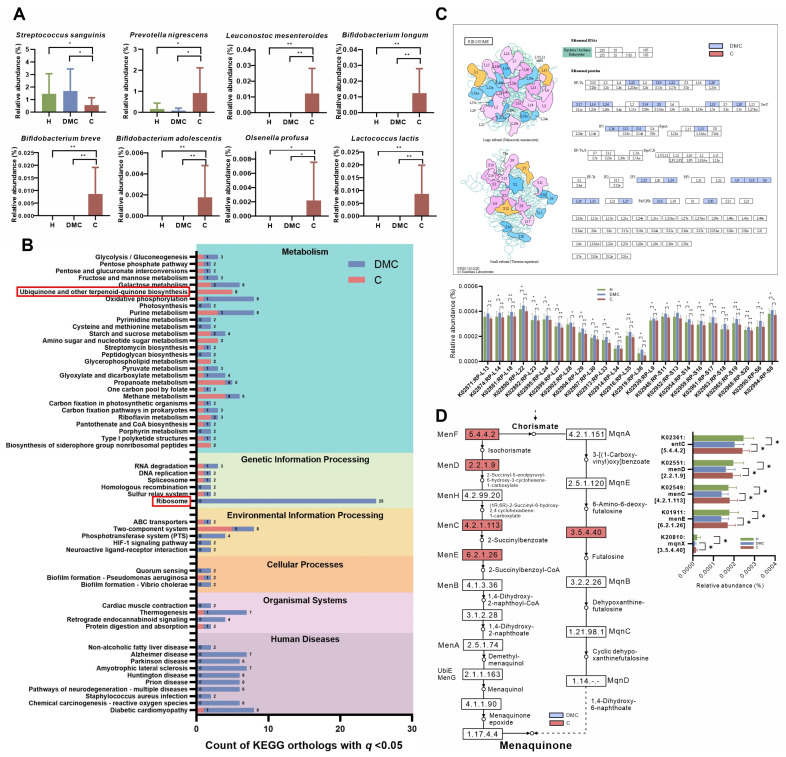
Taxonomic and functional differences in the microbiota between deciduous molars of the DMC and C groups. (**A**) Differentially abundant species. (**B**) KO counts differentially distributed in the KEGG pathways. Red boxes mark ribosome pathway, which had the most counts of KO and ubiquinone and other terpenoid-quinone biosynthesis pathways, which had a relatively large counts of KO, and, more importantly, KOs were all enriched in the deciduous molars of the C group. (**C**) Visualization and relative abundances of 25 differentially abundant KOs in the ribosome pathway. (**D**) Differentially abundant KOs in ubiquinone and other terpenoid-quinone biosynthesis pathways, particularly those linked to menaquinone biosynthesis. B, C, and D pathways were reconstructed using the KEGG mapper tool. * *q* < 0.05, ** *q* < 0.01.

**Figure 3 microorganisms-11-02461-f003:**
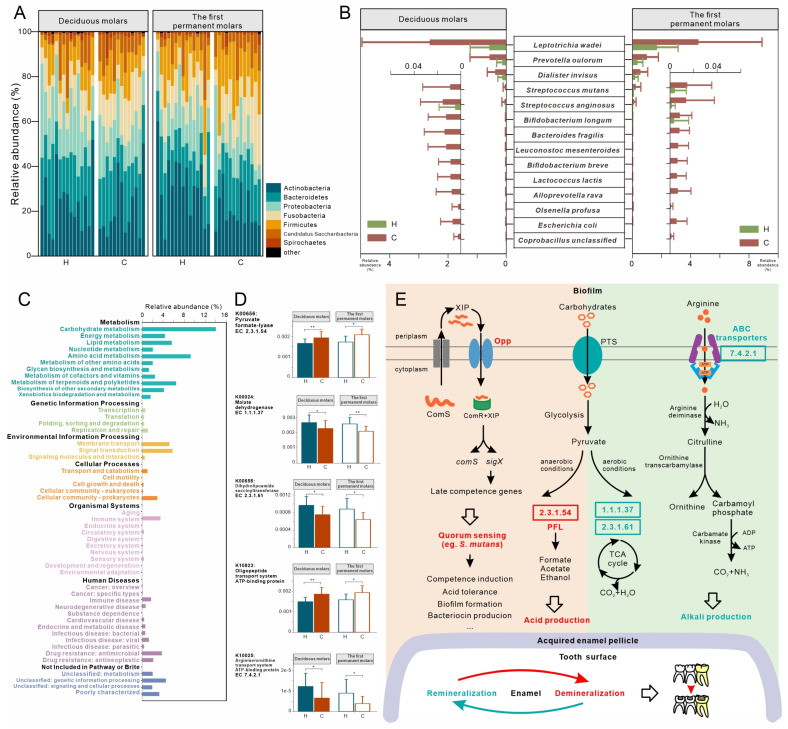
Taxonomic and functional characterization of plaque microbiota in the C group compared to healthy children (H group). (**A**) Relative abundances of major phyla in deciduous and first permanent molars. (**B**) Relative abundances of differentially abundant species overlapping between the two tooth locations. (**C**) The outline of KEGG level 2 pathways of H and C groups. (**D**) Relative abundances of KOs related to microbial anaerobic (K00656) and aerobic (K00024 and K00658) metabolism of carbohydrates, quorum sensing system (K10823), and transport of arginine (K10025). These KOs were differentially distributed between the H and C groups at deciduous and first permanent molars. (**E**) Schematic diagram showing potential pathways of plaque microbiota responsible for deciduous molar and adjacent permanent molar caries in C group children. Red text and background denote functions enriched in children with caries; green denotes functions enriched in healthy children. * *q* < 0.05, ** *q* < 0.01.

**Figure 4 microorganisms-11-02461-f004:**
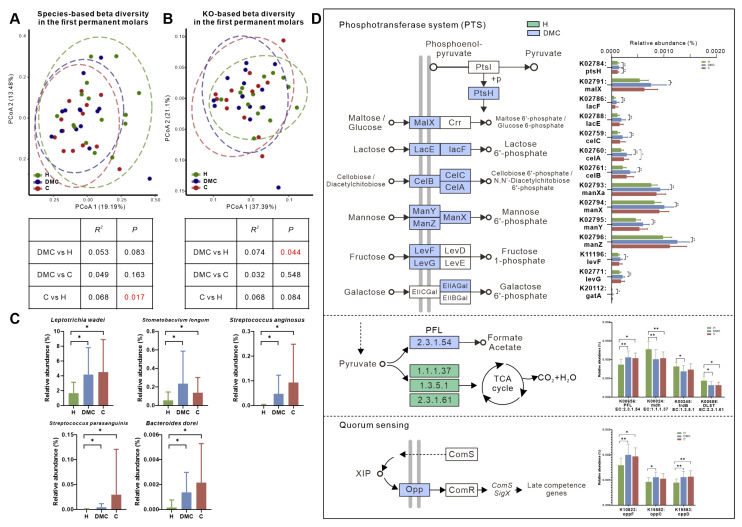
Microbial compositional and functional profiles of first permanent molars in the DMC group. (**A**,**B**). Overall microbial taxonomic and functional beta diversity in the first permanent molars of three groups. Principal coordinate analysis was conducted based on species- (**A**) and KO-level (**B**) Bray-Curtis distance; the nonparametric method Adonis (*n* = 999) was used to examine community differences among groups. Red text indicates *p* < 0.05. (**C**) Species that were differently distributed between the first permanent molars of DMC and H groups but not between DMC and C groups. (**D**) Differentially abundant KOs between the DMC and H groups related to PTS, pyruvate metabolism, and quorum sensing. * *q* < 0.05, ** *q* < 0.01.

**Table 1 microorganisms-11-02461-t001:** Demographic and clinical characteristics of the study population.

Group	Group Illustration	Sample Size	Gender ^a^(Male/Female)	Age ^a^(y)	dmfs (Deciduous Molars) ^b^*	dmfs (Deciduous Teeth) ^b^*	DMFS (First Permanent Molars)	DMFS (Permanent Teeth)
H	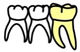	15	8/7	9.1 ± 1.4	0	0	0	0
DMC	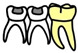	14	7/7	8.4 ± 1.3	10.1 ± 3.7	11.7 ± 4.3	0	0
C	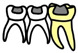	13	7/6	8.5 ± 1.1	15.2 ± 6.5	18.3 ± 9.3	3.8 ± 1.8	4.1 ± 2.3

Data are expressed as means ± standard deviation. dmfs/DMFS, decayed, missing, and filled surfaces. ^a^ Kruskal-Wallis test. ^b^ Mann-Whitney U test. * *p* < 0.05.

## Data Availability

The datasets analyzed in this study can be found in the NCBI Sequence Read Archive (SRA) database under BioProject PRJNA913442.

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
