# Peer review of "Microbial Relationship of Carious Deciduous Molars and Adjacent First Permanent Molars"

_microorganisms, 2023, doi:10.3390/microorganisms11102461_

Round 1

Reviewer 1 Report

The authors investigated the microbiological aspects of supragingival plaque in primary molars and adjacent first permanent molars in mixed dentition, evaluating the potential relationships of caries experiences in the examined primary and permanent molars.

Generally all parts of the manuscript are written clearly and correctly but some questions had been arised:

Materials and Methods:

Among the exclusion criteria: „ …fluoride treatment within the past 3 months” – What does it mean (e.g.: use of fluoride containing toothpaste? does it belong to this category?)? What is the importance of the 3 month criteria?This part might be completed.

Results:

In my opinion number of participants, age, gender should be involved rather in the „Materials (Patients) and Methods” („Study population”) but the Editor can decide in this question.

Nice figures demonstrate the results in details.

The final conclusions call the readers’ attention for the increased risk of caries in sound first permanent molars adjacent to carious primary molars. This finding is important for the clinical practice showing and strenghening the necessity of conservative treatment in primary teeth in case of mixed dentition.

I agree with the mentioned limitations, these are true, however, the study are performed correctly and gives useful, exact information. The applied methods are high level.

I suggest to publish the manuscript.

14/09/2023

Author Response

Materials and Methods:

Among the exclusion criteria: „ …fluoride treatment within the past 3 months” – What does it mean (e.g.: use of fluoride containing toothpaste? does it belong to this category?)? What is the importance of the 3 month criteria? This part might be completed.

 Response: Sorry for this inaccurate description. The exclusion criteria (vi) should be “antibiotic use or fluoride varnish application within the past 3 months”. We have revised the statements in the revised manuscript (Line 79-80). A previous study[1] indicated that fluoride varnish application had a significant disturbance effect on the microbial community structure of supragingival plaque in 3 days after fluoridation, and the microbial community recovered close to its original level 14 days after fluoride varnish application. Ready et al[2] demonstrated that the use of antibiotic amoxicillin whithin the 3 months prior to sampling significantly increased both the total number of amoxicillin-resistant bacteria and the proportion of the plaque microbiota resistant to amoxicillin. Given to the effect of antibiotic use and fluoride varnish application on the plaque microbiota, the exclusion criteria (vi) was applied.

References:

[1] Zhang Q, Guan L, Guo J, Chuan A, Tong J, Ban J, Tian T, Jiang W, Wang S. Application of fluoride disturbs plaque microecology and promotes remineralization of enamel initial caries. J Oral Microbiol. 2022 Jul 27;14(1):2105022. doi: 10.1080/20002297.2022.2105022. PMID: 35923900; PMCID: PMC9341347.

[2] Ready D, Lancaster H, Qureshi F, Bedi R, Mullany P, Wilson M. Effect of amoxicillin use on oral microbiota in young children. Antimicrob Agents Chemother. 2004 Aug;48(8):2883-7. doi: 10.1128/AAC.48.8.2883-2887.2004. PMID: 15273096; PMCID: PMC478491.

Results:

In my opinion number of participants, age, gender should be involved rather in the „Materials (Patients) and Methods” („Study population”) but the Editor can decide in this question.

Response: We have added the sample size of groups and age of participants in the “2.2. Study population and sample collection” (Line 85-87).

Nice figures demonstrate the results in details.

The final conclusions call the readers’ attention for the increased risk of caries in sound first permanent molars adjacent to carious primary molars. This finding is important for the clinical practice showing and strenghening the necessity of conservative treatment in primary teeth in case of mixed dentition.

I agree with the mentioned limitations, these are true, however, the study are performed correctly and gives useful, exact information. The applied methods are high level.

I suggest to publish the manuscript.

Response: Thank you for reviewing the paper thoroughly and giving constructive suggestions to help us improve the quality of the paper.

Reviewer 2 Report

Should the teeth have been separated with a matrix band during collection? Please explain and discuss.

The manuscript addresses the topic appropriately. The topic is relevant indeed and is well covered. The information elucidated is not extremely important, but it does fill a small gap in the knowledge base. It does in fact add to the knowledge base. There are very minor redactions required. The conclusions are consistent with the topics researched. Adding control cases to this type of work may be difficult and not required. The conclusions are consistent with the evidence presented and the posited question. The references are appropriate. The tables seem accurate according to the presented data. My question to the author/s is: why was there not a barrier between the teeth during collection. This may be a flaw in the data gathering aspect.

Author Response

My question to the author/s is: why was there not a barrier between the teeth during collection. This may be a flaw in the data gathering aspect.

Response: Thanks for this constructive suggestion! The teeth should have been separated with a band. We have added this to the limitation part (Line 360-361). A matrix barrier between the teeth could really improve the accuracy of sampling and will be applied in the follow-up studies.

Reviewer 3 Report

A very interesting paper and forms a foundation for further research. The metagenomic approach would be more informative than the 16S rRNA gene strategy - informs what metabolic processes are happening rather than what genera/species are present. Just a few comments/corrections for the authors to address:

1. Abstract (and main text, e.g. line 345) - what do the authors mean by "subtype like C" or "subtype like DMC"? Do they mean C-like subtype/DMC-like subtype?

2. What are scaftigs? Do they mean scaffolding contigs? Please define clearly on first use.

3. PCA = Principal coordinate analysis, not Principle.

4. Line 154 - state the range of Gb obtained across the samples.

5. Line 296 - "... C group lacked the presence of S. sanguinis..."

6. Line 347 - what is S. longum? Do the authors mean Bifidobacterium longum (B. longum)? They did not mention Bifidobacterium longum much in earlier sections of the text.

7. Line 358 - does not allow (do not use abbreviations like doesn't).

8. Line 359 - "... came before...". Came before what?

9. Line 362 - "Further work, such as a longitudinal study with a much larger sample size, is needed..." would be better wording.

No issues except for those highlighted in the Comments section.

Author Response

  1. Abstract (and main text, e.g. line 345) - what do the authors mean by "subtype like C" or "subtype like DMC"? Do they mean C-like subtype/DMC-like subtype?

Response: We apologize for this confusion. We have modified the statement in the Abstract, Line 323 and 348 in the revised manuscript.

  1. What are scaftigs? Do they mean scaffolding contigs? Please define clearly on first use.

Response: We have revised the statements in Line 128-129.

  1. PCA = Principal coordinate analysis, not Principle.

Response: We have revised the statements in Line 140.

  1. Line 154 - state the range of Gb obtained across the samples.

Response: The range of Gb obtained across the samples was (1.24 – 15.36 Gb). We have added it in Line 157.

  1. Line 296 - "... C group lacked the presence of S. sanguinis..."

Response: Thank you for pointing it out. We have revised it in the manuscript (Line 300).

  1. Line 347 - what is S. longum? Do the authors mean Bifidobacterium longum (B. longum)? They did not mention Bifidobacterium longum much in earlier sections of the text.

Response: S. longum mentioned in Line 347 was abbreviation for Stomatobaculum longum. It was first mentioned with its full name in the Abstract (Line 27).

Reference: Sizova MV, Muller P, Panikov N, Mandalakis M, Hohmann T, Hazen A, Fowle W, Prozorov T, Bazylinski DA, Epstein SS. Stomatobaculum longum gen. nov., sp. nov., an obligately anaerobic bacterium from the human oral cavity. Int J Syst Evol Microbiol. 2013 Apr;63(Pt 4):1450-1456. doi: 10.1099/ijs.0.042812-0. Epub 2012 Jul 27. PMID: 22843721; PMCID: PMC3709536.

  1. Line 358 - does not allow (do not use abbreviations like doesn't).

Response: We have revised this statement (Line 362).

  1. Line 359 - "... came before...". Came before what?

Response: We apologize for the confusion generated by the previous version of the manuscript. We have revised the statement as follows: ‘the fact that the cross-sectional design of the study does not allow for the conclusions such as the higher abundance of cariogenic species in the deciduous molar do not necessarily mean the child will develop caries in the permanent molar;’ (Line 361-364).

  1. Line 362 - "Further work, such as a longitudinal study with a much larger sample size, is needed..." would be better wording.

 Response: Thank you for the suggestion. We have revised the statement as suggested above (Line 367).